

# Fluvial fans in the Himalayan Terai: A gradual shift of the Karnali River from double to single branch

Kshitiz Gautam[1], Astrid Blom[1], Mathieu Roebroeck[1], Marijn Wolf[1], and Thom Bogaard[1]

[1]Faculty of Civil Engineering and Geosciences, Delft University of Technology, Netherlands

**Correspondence:** Kshitiz Gautam (k.gautam@tudelft.nl)

**Abstract.** Fluvial fans in the Himalayan Terai are essential for water resources and provide crucial habitats for endangered species, including tigers. Switching of the dominant channel in these fans influences such habitats by changing the distribution of water and sediment. This study addresses such a transition in the Karnali River, one of the least human-altered large rivers in Nepal and India. For over two centuries, the Karnali maintained a double-branch system, but in recent years it has gradually consolidated into a single branch. Our primary objective is to describe this shift and to assess its trigger. By analyzing flow duration curves, fluvial fan topography, and channel properties, we suggest the cause is an extreme monsoon season in 2009, when two major peak discharges seem to have initiated the subsequent gradual deposition of coarse sediment at the upstream end of the eastern branch (Geruwa), effectively gradually plugging its flow. To better understand the balance between natural and anthropogenic influences, we compare the Karnali with the more heavily altered Koshi River. While embankments and infrastructure developments have significantly shaped the Koshi's morphology, the Karnali's shift appears to be driven primarily by natural sediment dynamics. Human interventions (such as embankments and existing hydropower dams) appear to have played little to no role in the transition. With rapid hydropower expansion and ongoing modifications to the river system, we anticipate that Karnali's single-channel configuration will persist, with profound implications for water distribution and habitat conservation in Bardiya National Park.

## 1 Introduction

Fluvial fans dominate the Himalayan foreland zone, and form where rivers exit the mountain chain and deposit large volumes of sediment (DeCelles and Cavazza, 1999; Geddes, 1960; Leier et al., 2005). Unlike alluvial fans, fluvial fans are integral to a broader river system, with channels that distribute sediment across a wider area and maintain more consistent flow over time (DeCelles and Cavazza, 1999; Geddes, 1960; Hansford and Plink-Björklund, 2020; Leier et al., 2005; Quick et al., 2023; Sinha and Friend, 1994). In western Nepal and India, these fans shape the Terai Arc Landscape, while in eastern India and Bhutan, they are known as Duars (Thapa and Tuladhar, 2021; Wikramanayake et al., 2010). Channel avulsions are a common feature of fluvial fans (Dingle et al., 2016; Sinha et al., 2005; Slingerland and Smith, 2004; Parker et al., 1998), often leading to sudden and large shifts in channel course. For example, the Koshi River in the Terai has experienced a channel course change of over 100 km in the past two centuries (Agarwal and Bhoj, 1992; Chakraborty et al., 2010). These dramatic shifts are often triggered by peak flow events or have a more stochastic nature (Wells and Dorr, 1987).





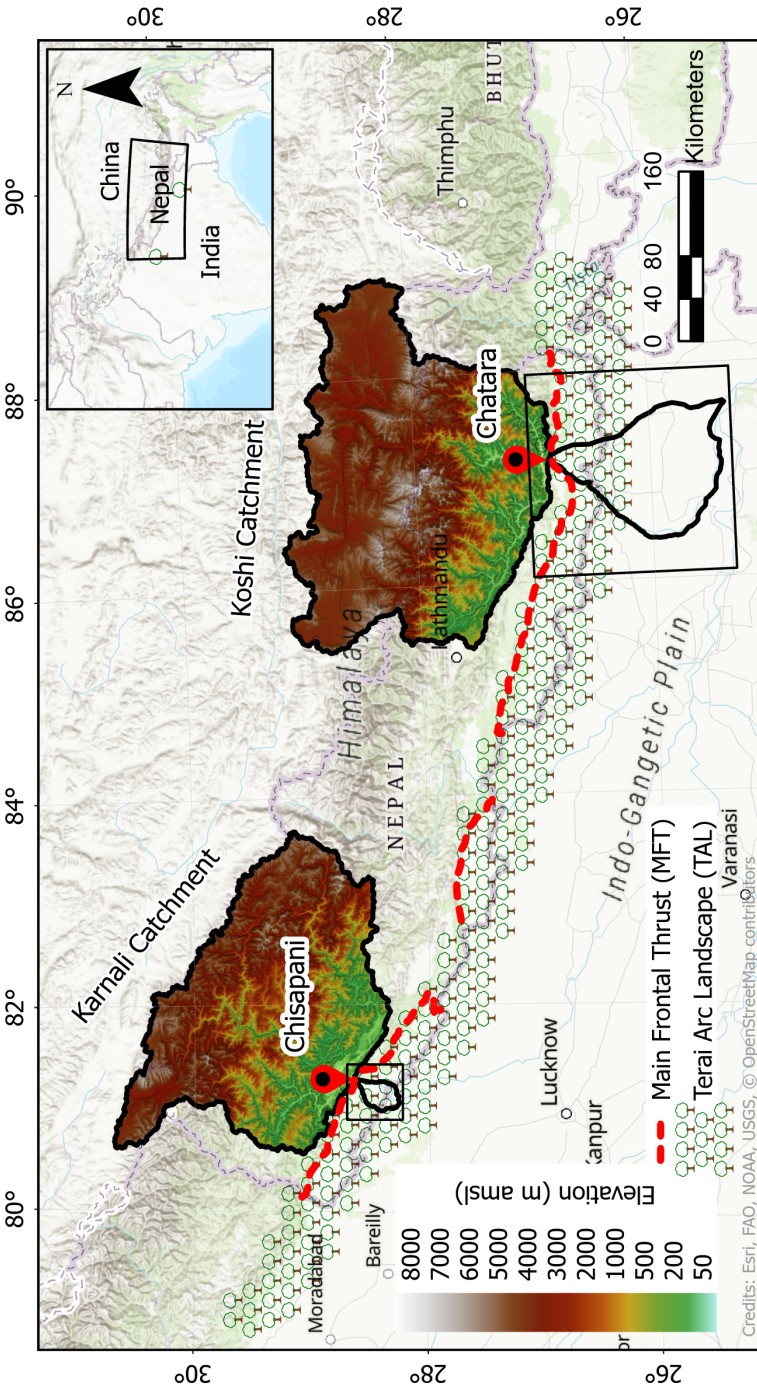

**Figure 1.** Catchment areas of the Karnali and the Koshi river systems in the Terai, (upstream of, respectively, Chisapani and Chatara, where the rivers exit the Himalayan mountain chain) are of comparable size. The Karnali fluvial fan is ten times smaller than the Koshi fluvial fan.





Rivers in the Terai are crucial for both sustaining wildlife habitats that are home to species like the Bengal tiger (Chanchani et al., 2014; Wikramanayake et al., 2010, 1998), one-horned rhinoceros (Ghimire, 2020; Thapa et al., 2013), gharial crocodile (Vashistha et al., 2021), and elephants (Thapa et al., 2019), and supporting agriculture in the fertile Terai (Thapa and Tuladhar, 2021). Balancing these vital functions requires the sustainable management of Terai rivers and their surrounding landscapes.

The Karnali fluvial fan (Rakhal et al., 2021; Dingle et al., 2020a) is part of the Ghagra plain, which is a subregion within the Ganga Plain (Dingle et al., 2017, 2016; Goswami and Mishra, 2013; Lupker et al., 2012). After exiting the Himalayan mountain chain at Chisapani (Figure 1), the Karnali River splits into two main branches (Rakhal et al., 2021): the eastern Geruwa branch and the western Kauriala branch. The channel bed transitions from being gravel-dominated to sand-dominated (Dingle et al., 2021; Blom et al., 2017) along each of the two branches (Dingle et al., 2020b, 2017). This abrupt gravel-sand transition is
located where the sediment flux transported from upstream runs out of gravel and a gravel front progrades (Blom et al., 2017; Dingle et al., 2017; Parker and Cui, 1998; Paola et al., 1992). The two branches rejoin forming a confluence approximately 50 km downstream of the bifurcation, just upstream of the Kailashpuri Dam in India. The Geruwa branch flows through Bardiya National Park, a critical habitat for endangered wildlife, including tigers (Thapa and Tuladhar, 2021; Wikramanayake et al., 1998).

Anthropogenic interventions are an important stressor on river systems in the Terai (Swarnkar et al., 2021; Bajracharya et al., 2018; Haddeland et al., 2014; Lutz et al., 2014; Magilligan and Nislow, 2005). Dams increase low flows and reduce high flows, diminishing peak flows and overbank events. This alters connectivity between the main channel and floodplain (Magilligan et al., 2003). Furthermore, dams trap approximately 50% of global river sediment, leading to reduced sediment delivery and downstream channel incision (Vörösmarty et al., 2003; Schmidt and Wilcock, 2008). For instance, the narrowing
of the Yamuna River in the western Terai has been linked to upstream dams and sediment mining since 2013 (Yadav et al., 2023).

Climate change intensifies stress on Terai river systems. Rising temperatures and increased moisture lead to heavier rainfall, while earlier snow melt in the surrounding mountains increases runoff. This results in higher river flows, more frequent flooding, and prolonged peak events, with serious impacts on agriculture and biodiversity (Bajracharya et al., 2018; Immerzeel
et al., 2010; Lutz et al., 2014; Oki and Kanae, 2006). These effects are especially pronounced in Terai basins already impacted by human activities like damming and land-use changes (Palmer et al., 2008).

This study investigates the shift of the Karnali River from a double-branch to a single-branch system over the past two decades. Discharge in the eastern branch has steadily declined, as well as the flow into a major irrigation inlet at Okhariya, located between the original branches. Recent sediment excavation at the upstream end of the eastern branch and irrigation inlet
has resulted in only a minor increase in Geruwa flow. The western branch (Kauriala) now carries the majority of the Karnali River's flow, impacting both the ecosystem and wildlife. This shift has challenged Bardiya National Park, where the reduced flow in the Geruwa branch has led to habitat changes, promoting the growth of shrub and woody vegetation (Vashistha et al., 2021). Additionally, Ganges River dolphins have been displaced from the Geruwa branch, which is a protected area free from fishing, due to the altered flow dynamics (Vashistha et al., 2021; Khanal et al., 2016).





The objective of this study is to examine the transition of the Karnali River from a double-branch to a single-branch system, exploring whether these changes are driven by anthropogenic interventions, climate change, or natural factors. We analyze catchment characteristics, hydrograph data, and flow duration curves. In addition, we compare the Karnali River to the Koshi River (Wells and Dorr, 1987; Gautam and Acharya, 2012), which is more heavily impacted by human activities. Our methodology includes using a Digital Elevation Model (DEM) to assess elevation within the catchments and fluvial fans, alongside historical maps and satellite imagery to investigate changes in the fluvial fan and channel dynamics. The paper is structured as follows: Section 2 outlines our methods, Section 3 presents the flow characteristics of the two catchments, Section 4 examines the dynamics of the fluvial fans, and Section 5 assesses channel properties such as sinuosity and braiding.

## 2 Methods

Here we explain our methods to analyze and compare (a) catchment and hydrological characteristics, (b) fan elevation and dynamics, and (c) channel sinuosity and braiding in the Karnali and Koshi river systems. Two locations are key to our analysis (Figure 1): Chisapani, where the Karnali River exits the Himalayan mountain range, and Chatara, where the Koshi River transitions to the flatlands of Nepal and India.

We assess the hydrological behavior of both river systems by analyzing the measured discharge time series from Chisapani (1962-2019) and Chatara (1977-2015).

We define the area-averaged river discharge ($q_A$) as the ratio of the measured water discharge or outflow at Chisapani and Chatara ($Q$) to the associated upstream catchment area ($A$). We compare flow duration curves, daily discharge, monthly-averaged discharge, and percentile discharges. Specifically, we analyze temporal trends of discharge values exceeded for 10%, 50%, and 90% of the year, as minimum and maximum values are more susceptible to measurement errors.

To evaluate the impact of hydropower on discharge trends in both catchments, we compare periods before and after 2000, when hydropower in the Koshi catchment significantly expanded. In 2000 only five hydroelectric plants (larger than 1 MW capacity) were operational in the Koshi catchment according to data available from the Nepal Department of Electricity Development. By 2022 this number had increased to 28, and 76 plants with a license for construction, and 132 more under feasibility study. In contrast, the Karnali catchment remains relatively undeveloped (Shrestha and Paudyal, 1992). By 2022, it had only two operational hydropower plants, with 12 in construction, and 30 additional projects under feasibility study.

We use the ALOS World 3D 30m-resolution elevation (AW3D30) data set to compute catchment area, average slope, stream frequency, and drainage density. For fan elevation analysis, we use the 30m-resolution TanDEM-X data set. Fan outlines and former channel courses are delineated using literature (Rakhal et al., 2021; Singh, 1971; Wells and Dorr, 1987), historical maps, and Landsat 4, 5, 7, and 8 satellite images. Maps of the Karnali system span 1785–1967, though those predating 1855 have coarse spatial resolution and cannot be georeferenced, yet help identify historical flow paths. The channel course since 1760 and maps of the Koshi system are derived from Chakraborty et al. (2010). We digitize channel courses from maps and satellite images using ArcGIS Pro 3.1.3.



To analyze deviations from the mean elevation decline along the fan, we define normalized elevation as the elevation along each cross-section minus the minimum elevation within that cross-section. Cross-sections are taken perpendicular to the fan's length from apex to toe, with a spacing of 500 m for the Karnali fan (50 km length) and 1600 m for the Koshi fan (160 km length).


We use 30m-resolution Landsat images (1972–2021) to estimate flow partitioning between the two branches of the Karnali system. Our analysis focuses on a 25 km reach downstream of the Chisapani bifurcation and upstream of tributary confluences with the Kauriala and Geruwa branches. To quantify water cover, we select the first low-flow image of each year with minimal cloud cover. We apply the Normalized Difference Water Index (NDWI) (McFeeters, 1996) to identify water-covered pixels

in each branch, allowing us to compute the percentage of water cover. This percentage serves as a rough proxy for the flow partitioning between the two branches. The irrigation inlet between the Kauriala and Geruwa branches, with a maximum capacity of 50 m$^3$/s, is minor relative to the main channel and thus excluded from the flow partitioning analysis.

We assess braiding and sinuosity of the Karnali River between 1990 and 2015 using Landsat 5, 7, and 8 imagery captured during low-flow conditions to better observe channel dynamics. We quantify changes using dimensionless indices: the Braiding Index ($B_I$) and Sinuosity Index ($S_I$) following Friend and Sinha (1993), and the Sinuosity Dynamics ($S_D$). The Braiding Index,

$B_I$, is defined as the ratio of the total channel length ($L_{Ctot}$) to the mid-channel length of the widest channel ($L_{Cmax}$):

$$B_I = \frac{L_{Ctot}}{L_{Cmax}} \qquad (1)$$

The Sinuosity Index, $S_I$, is defined as the ratio of mid-channel length of the widest channel, $L_{Cmax}$, to the linear length of the segment of the river under consideration, $L_R$:

$$S_I = \frac{L_{Cmax}}{L_R} \qquad (2)$$

Sinuosity Dynamics, $S_D$, is defined as the percentage of change of the Sinuosity Index in year $n$ over two consecutive years:

$$S_D = \frac{S_I^{n+1} - S_I^n}{S_I^n} \times 100\% \qquad (3)$$

We subdivide the Karnali system into segments of approximately equal length (Figure 2), and compute the values of the three indices for each segment.

We use NDWI images to determine the length of the widest channel and total length of the channels. NDWI images allow one to distinguish the widest flow-filled channel during low-flow periods. We term this widest channel the main channel, the length of which enables us to compute the braiding and sinuosity indices. We distinguish other channels, either less wide or dry-bedded, through visual interpretation of the true color composition to determine the total length of the channels in the considered segment.





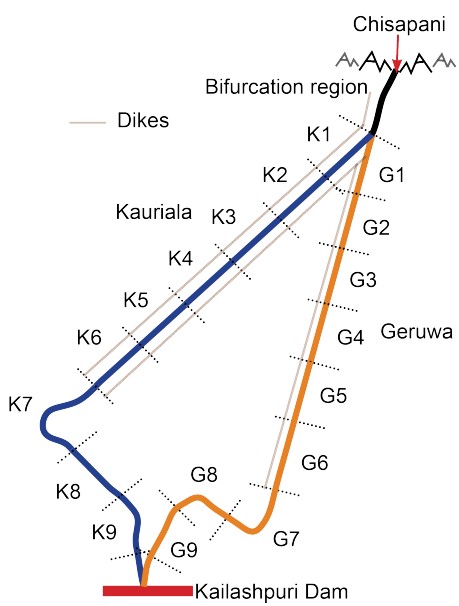

**Figure 2.** The two Karnali branches are divided into nine segments, each about 5 km long. Blue sections represent the Kauriala branch, and orange ones the Geruwa branch. For each segment, we compute three indices: the Braiding Index, Sinuosity Index, and Sinuosity Dynamics.

## 3 Catchment and Hydrology

We have determined catchment characteristics in terms of catchment area, average slope, stream frequency, and drainage density (see Table 1). The catchment areas of the Karnali River (upstream of Chisapani) and the Koshi River (upstream of Chatara) are approximately 45,500 km$^2$ and 57,800 km$^2$, respectively, implying that they are of the same order of magnitude.

**Table 1.** Catchment characteristics of the Karnali River (upstream of Chisapani) and Koshi River (upstream of Chatara).

| Catchment Characteristics | Karnali | Koshi |
|---|---|---|
| catchment area | 45,500 km$^2$ | 57,800 km$^2$ |
| elevation range | 170-7750 m amsl | 90-8800 m amsl |
| average slope | 63% | 45% |
| stream frequency | 0.6 streams/km$^2$ | 0.2 streams/km$^2$ |
| drainage density | 0.67 km/km$^2$ | 0.45 km/km$^2$ |

Elevation of the Karnali River catchment varies from nearly 7750 m amsl to 170 m amsl at Chisapani, and that of the Koshi River varies from nearly 8800 m amsl to 90 m amsl at Chatara. About 17% of the Karnali River catchment and 28% of the Koshi River catchment is above 5000 m amsl. The Karnali catchment is steeper than the Koshi catchment, with a 63% versus a





45% mean slope. We find that the Karnali catchment has a higher stream frequency than the Koshi, with 0.6 versus 0.2 streams per square kilometer, as well as a higher drainage density (0.67 km/km$^2$ versus 0.45 km/km$^2$ for Koshi).

**Figure 3.** Daily-average, monthly-average, maximum and minimum area-averaged discharge, $q_A$, for (a) the Karnali system and (b) the Koshi system. Flow duration curves for the area-averaged discharge, $q_A$, for (c) the Karnali and (d) the Koshi system. 10, 50, and 90 percent values of the area-averaged discharge ($q_{A10}$, $q_{A50}$, and $q_{A90}$) for (e) the Karnali and (f) the Koshi system.



Seasonal discharge variation is similar between the two river systems (Figure 3a,b). Water discharge is low during winter,
between December to April. Water discharge increases due to glacial melt driven by temperature increase starting from late
spring. Monsoon precipitation further increases the water discharge from June to September, providing a high-flow period
typical of monsoon-dominated river systems.

The Karnali system has a shorter high-flow period and a slightly higher monthly-averaged flow rate in August than the
Koshi system. This can be attributed to the amount and duration of rainfall these systems receive: the eastern Koshi catchment
receives a larger amount of rainfall for a longer period compared to the western Karnali catchment (Bookhagen and Burbank,
2010; Geddes, 1960; Khatiwada et al., 2016; Sinha et al., 2005). The Karnali's peak discharge in August occurs one month
later than the peak precipitation (July) over the basin (Khatiwada et al., 2016).

The maximum daily discharge varies more strongly in the Karnali River than in the Koshi system (Figure 3a,b). This higher
variability can be attributed to the Karnali catchment's larger steepness, stream frequency, and drainage density, as these
characteristics shorten the run-off response time to rainfall.

The flow duration curve of the Karnali system seems reasonably constant (Figure 3c,e). The area-averaged discharge in
the Koshi system seems to have decreased since 2000 (Figure 3d,f). The latter can be attributed to the rapid development of
hydropower in the Koshi system since 2000 (Figure 4). A similar decrease in high and moderate flows due to dam construction
has been reported for US river systems (Magilligan et al., 2003).

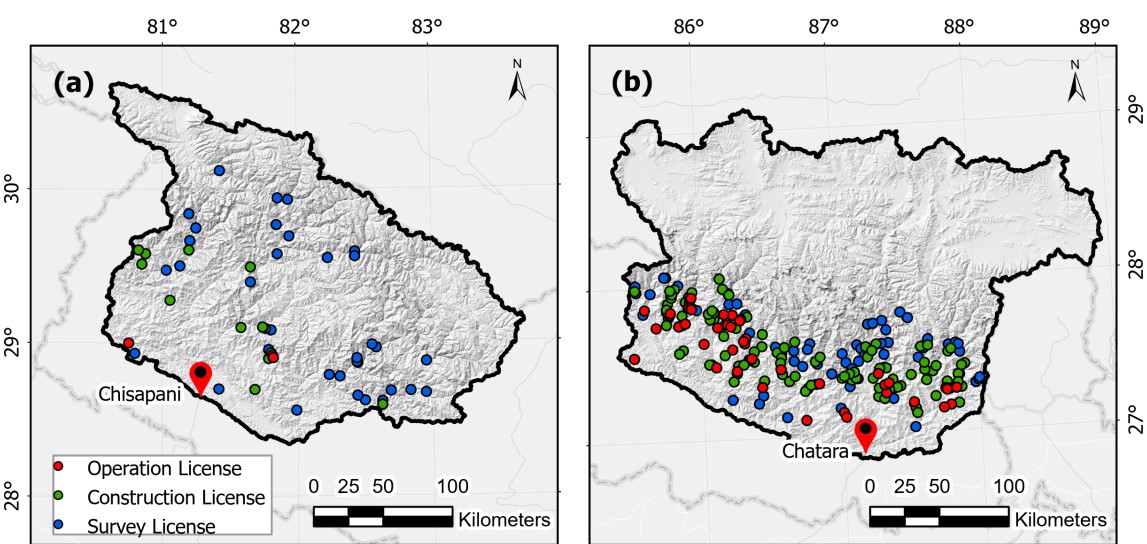

**Figure 4.** Hydropower stations operational as of 2022 (red dots), projects with a construction permit (green dots), and projects under feasibility study (blue dots) in (a) the Karnali catchment and (b) the Koshi catchment.



## 4   Fan Dynamics

The Koshi fan is ten times larger than the Karnali fan (Figure 5a,b): the Karnali fan has an area of about 1000 km$^2$ and the Koshi fan has an area of about 10000 km$^2$. This difference in fan size may be due to a difference in sediment yield from the upstream catchment and geological constraints. Sediment yield of the Karnali at Kailashpuri Dam (B.K. Ghat station) is approximately 1456 t/km$^2$/year or 64 Mt/year (Sinha et al., 2005), whereas the sediment flux in the Koshi River at Chatara equals 1915 t/km$^2$/year or 101 Mt/year (Sinha et al., 2023, 2019). The higher sediment yield of the Koshi basin shows its potential to deposit larger quantities of sediment in its fan than the Karnali. Dingle et al. (2016) find that the basins in the west subside more slowly than in the east, implying that the smaller Karnali fan area is not explained by a higher subsidence rate. Furthermore, they note that the lower subsidence rates in the west are associated with entrenched river systems.

Figure 5 shows that the Karnali River has functioned as a two-branch system since at least 1785, with dominance shifting between the branches. In contrast, the Koshi River has remained a single-branch system, though its main channel has migrated over time (Chakraborty et al., 2010). The stability of river bifurcations remains a debated topic: some researchers argue that bifurcations with unequal downstream branches can reach a stable equilibrium (Edmonds and Slingerland, 2008), while others suggest that bifurcations are inherently unstable (Kleinhans et al., 2013). Other studies propose that multiple stable equilibrium states are possible in bifurcation systems (Schielen and Blom, 2018).

Our earliest map of the Karnali system dates to 1785, showing Geruwa as the dominant branch. By 1861, 1893, and 1906, the Kauriala branch had become the larger of the two (Figure 5a). From 1933 onward, Geruwa resumed dominance. The most recent shift followed a double-peaked monsoon in 2009, after which Kauriala's flow rate gradually increased. NDWI analysis of satellite images confirms that since 2009, the water-covered area of the Kauriala branch has gradually grown at the expense of the eastern Geruwa branch (Figure 6a).

Both fluvial fans exhibit a typical decline in elevation along their length (Figure 5c,d). A geological constraint along the eastern boundary of the Karnali fan is marked by a sharp rise in elevation (Figure 5c), while the Koshi fan shows an elevation difference along its southern boundary (Figure 5d). These boundaries appear to have restricted the fans' expansion in these directions (Figure 5a,b).

Normalized elevation (in Figure 5e,f) illustrates deviation in local topography from the typical elevation decline along the fan (Section 2). It is defined as the elevation along each cross section minus the minimum elevation within that cross section, with cross-sections taken perpendicular to the fan's length from apex to toe. This analysis reveals that the central part of the Koshi fan is elevated (Figure 5f), resulting in a convex-up cross-sectional profile (Chakraborty et al., 2010). Such profiles restrict lateral channel migration through bank erosion, making avulsion the dominant mechanism for channel course changes.

Figure 5f suggests that a monsoon-driven avulsion just downstream of Chatara could redirect the channel toward the eastern part of the fan. This is supported by the 2008 breach in the eastern embankment and the subsequent avulsion upstream of the Koshi Barrage (Dingle et al., 2016). During this event, the Koshi temporarily reoccupied its paleochannels and floodplains until the embankment was repaired in 2009 (Sinha et al., 2014) .





**Figure 5.** Course of branches over (a) the Karnali fan and (b) the Koshi fan at various times. Solid lines indicate dominant branches and dashed lines represent smaller branches. Absolute elevation over (c) the Karnali fan and (d) the Koshi fan. Normalized elevation for (e) the Karnali fan and (f) the Koshi fan shows deviation from the mean elevation decline along the fan.





Similarly, though less pronounced, the central part of the upper Karnali fan is slightly elevated compared to its two branches (Figure 5e), which was indicated by Dingle et al. (2016) to be the result of its low subsidence rate. This elevation difference,
possibly aided by current embankments, limits flow from occupying the central part of the fan.

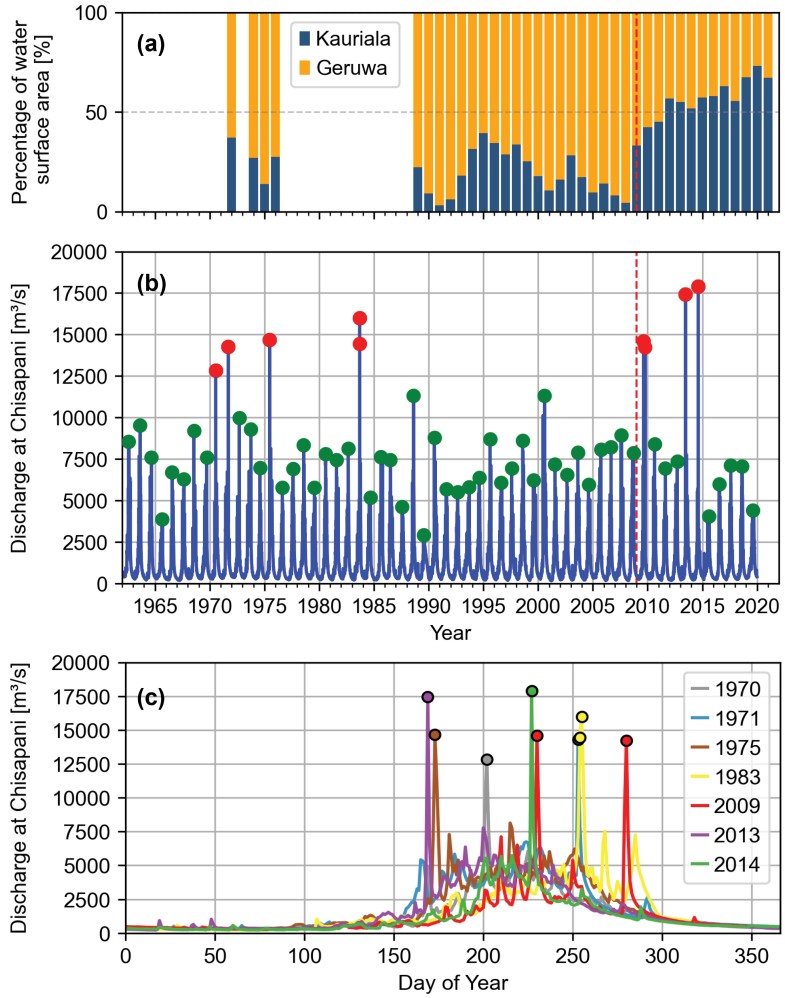

**Figure 6.** Water-surface cover and hydrograph characteristics in the Karnali system: (a) relative water-surface cover for the Kauriala and Geruwa branches of the Karnali River, as obtained from satellite images under low-flow conditions according to Roebroeck (2022); (b) hydrograph at Chisapani with red circles indicating peak flow values over 12500 m³/s and green circles annually-maximum values; and (c) annual hydrographs for years with high monsoon-driven peak flows over 12500 m³/s.

## 5 Channel Dynamics

Here we investigate the channel dynamics of the two branches of the Karnali River to explore a potential link to its gradual transition from a double-branch to a single-branch system since 2009.



The Kauriala branch is embanked on both its western and eastern sides, limiting floodplain area and restricting channel
dynamics and migration. The Geruwa branch remains mostly unembanked along its eastern bank, where it borders Bardiya
National Park. This absence of embankment has preserved the Geruwa's dynamic character.

Figure 7 shows Sinuosity and Braiding Indices and Sinuosity Dynamics for the segments of the two branches (Figure 2). We
observe that sinuosity does not significantly vary spatially (Figure 7a). Sinuosity change in Segment 7 of the Kauriala branch
is relatively high where it migrates laterally in response to a tributary (Figure 7b).

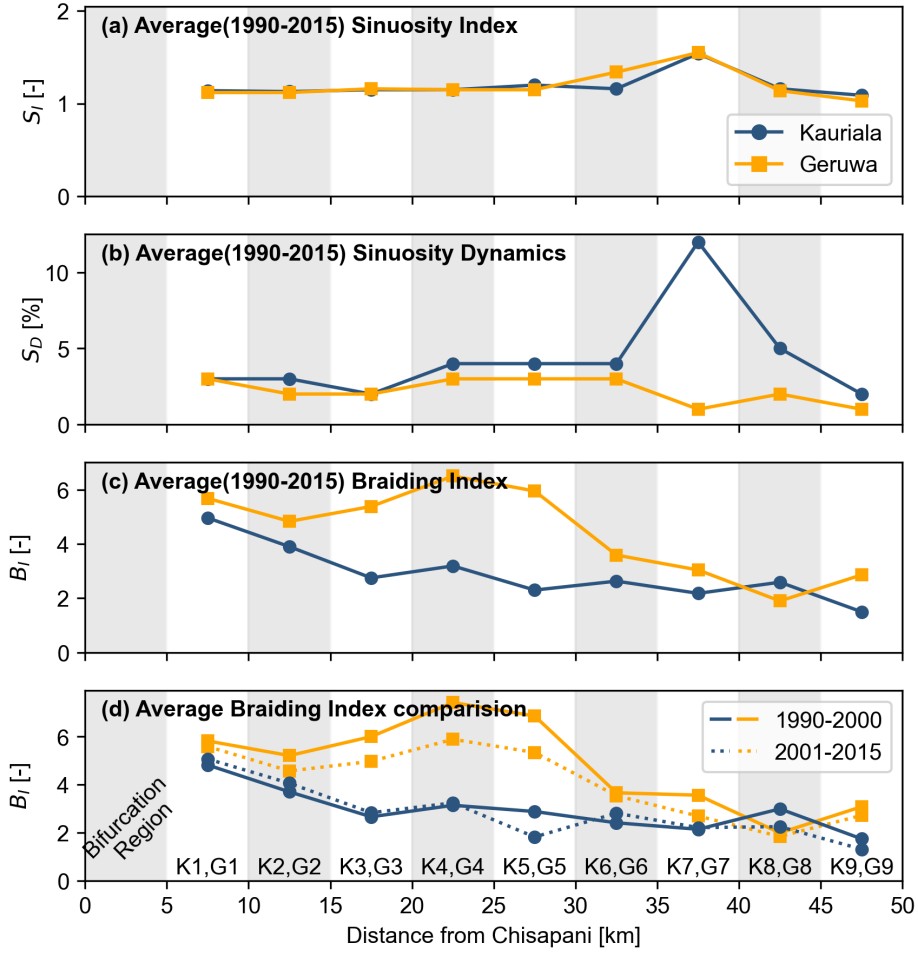

**Figure 7.** Sinuosity and braiding indices for segments of the Kauriala and Geruwa branches over the fluvial fan of the Karnali River:
(a) Sinuosity Index, (b) Sinuosity Dynamics, and (c) Braiding Index, all averaged over the period 1990-2015. (d) Braiding Index for two
periods, 1990-2000 and 2001-2015. Segments are defined in Figure 2, and river km starts at Chisapani with a dynamic bifurcation region
right downstream of Chisapani.



190 Braiding intensity decreases downstream (Figure 7c), with the highest values near the bifurcation, where channel and flood-plain width span several kilometers (see also Figure 8). This ample space allows for active channel migration and switching. The downstream decline in braiding intensity is largely due to embankments (Kafle, 2021; Venkateswaran et al., 2015) along the central and lower fan. Additionally, backwater effects from Kailashpuri Dam likely reduce channel dynamics in the lower reaches.

195 The braiding intensity of Geruwa generally exceeds that of Kauriala (Figure 7c), likely due to the absence of embankments along Geruwa's eastern bank. Over time, the braiding intensity of the Geruwa branch appears to decline slightly (Figure 7d), which may be linked to its decreasing share of upstream flow.

 Figure 8 shows the variation in bed surface grain size in the Karnali bifurcation region. Notably, Location 3 near 'Dolphin Point', which is an outer bend where the Geruwa branch originates, contains exceptionally coarse sediment, ranging from

200 coarse gravel to boulders up to 50 cm in diameter. The coarse bed sediment at the outer bend likely results from bend sorting (Baar et al., 2018; Parker and Andrews, 1985). In this process, the spiral flow and transverse slope in a river bend direct coarse sediment toward the outer bend, forming its bed surface, while finer sediment is transported preferentially toward the inner bend. When a downstream channel takes off from an outer bend, the bend sorting mechanism promotes the transport of coarse sediment into this channel (Chowdhury et al., 2023; Frings and Kleinhans, 2008). If the sediment is too large to remain mobile,

205 it deposits in the upstream reach of the bifurcate (Chowdhury et al., 2023).

 In summary, we hypothesize that the heavy 2009 monsoon season formed a sediment 'plug' in the upstream reach of the Geruwa branch, consisting of coarse sediment transported into the branch due to upstream bend sorting in the Karnali River. This plug appears to have initiated the gradual reduction in flow into the Geruwa branch since 2009.





**Figure 8.** Impression of spatial variation of bed surface grain size over the Karnali bifurcation region.





## 6 Discussion

An important question is whether the Karnali fluvial fan is evolving toward a single-branch system, similar to the Koshi River. This is particularly relevant because the Geruwa branch borders and supplies freshwater to Bardiya National Park, a key habitat for endangered species such as the tiger. Our historical analysis reveals frequent shifts in channel dominance between the Kauriala and Geruwa branches. Key factors driving these dynamics include grainsize-specific sediment supply and sediment transport capacity in the upstream reaches. Our findings suggest that sediment entering the Geruwa branch is either too coarse

or too abundant to be effectively transported downstream, leading to deposition and gradual closure. Similar mechanisms have been observed in the gradual shift of flow partitioning in the Rhine River (Blom et al., 2024; Chowdhury et al., 2023). The ongoing decline in Geruwa's discharge shows no signs of reversing.

We do not expect the shift in flow partitioning between the Kauriala and Geruwa branches and gradual closure of the Geruwa branch to be driven by human interventions, as hydropower stations yet remain scarce in the Karnali basin. While both branches

are embanked, except along the eastern bank of the Geruwa, these embankments do not appear to have influenced the recent shift in flow partitioning. However, in the absence of embankments, a gradually clogged branch might have led to the formation of one or more new channels (i.e., an avulsion) across the central part of the Karnali fluvial fan.

Over the past century, several major earthquakes have significantly impacted the Terai region, including the Bihar-Nepal Earthquake (1934), Udaypur Earthquake (1988), and Gorkha Earthquake (2015). These seismic events influence sediment

yield by triggering landslides, destabilizing hillslopes, and altering river channel morphology, often leading to a substantial increase in sediment supply (Marc et al., 2019; Lin, 2018). Earthquake-induced landslides can introduce large volumes of boulders and gravel into river systems (Marc et al., 2019). While fine sediment is typically flushed out quickly, coarse landslide deposits can persist in river valleys for decades to millennia before being gradually reworked and transported downstream, reshaping river morphology over time.

While some studies suggest that earthquakes enhance coarse sediment yield to rivers (Dadson et al., 2004; Hovius et al., 1997), others argue that particle abrasion during transport leads to an increased sand load downstream rather than a sustained increase in coarse material (Dingle et al., 2017). The above earthquakes may have contributed to a temporal coarsening and increased sediment flux at Chisapani and the Karnali bifurcation region compared to previous decades. Combined with the double-peaked 2009 monsoon season, this process may have initiated the gradual closure of the Geruwa branch.

An interesting question is whether natural processes or human interventions could eventually reverse this trend. As flow into the Geruwa branch declines, an increasing share of the coarse sediment flux will be diverted into the Kauriala branch. This sediment may deposit in the upstream reach of the Kauriala branch, which would gradually raise its bed level. Over time, this process could redirect flow back into the Geruwa branch, potentially reversing the current trend. If existing and future embankments are well maintained, these changes are unlikely to trigger an avulsion like that of the Koshi River but may

instead lead to the gradual re-opening of the Geruwa branch.

Human interventions aimed at reversing the ongoing decline in water discharge within the Geruwa branch could focus on modifying the bend configuration upstream of (or at) Dolphin Point (near Location 3 in Figure 8). This approach warrants





further investigation. However, given the intense dynamics of the Karnali River during the monsoon season, controlling its flow remains challenging, and the effectiveness of such interventions is highly uncertain.

With the planned hydropower stations and other interventions in the upstream Karnali catchment, we expect the likelihood of dominant channel switching to decrease. These stations will likely reduce variability in both water discharge and sediment flux. Furthermore, reservoirs trap coarse sediment more effectively than fine sediment, potentially resulting in a finer sediment load exiting the Himalayan mountain range. As a result, these changes will dampen flow and sediment dynamics, reducing the probability of a switch in dominant channels.

## 7 Conclusions

Over the past 15 years, the Karnali River in the Himalayan Terai has shifted from a double-branch to a predominantly single-branch system, despite maintaining a double-branch configuration since at least 1785.

Human interventions alone do not explain this change: embankments have only marginally reduced braiding and sinuosity, and hydropower dams are not yet widespread in the region.

We hypothesize that the shift is linked to the 2009 monsoon season, which featured two peak discharge events. Since then, the western Karnali branch (Kauriala) has increasingly dominated flow.

The double-peaked 2009 monsoon likely caused coarse sediment deposition at the upstream end of the eastern branch (Geruwa), which may have initiated a gradual reduction in its flow. This self-reinforcing process is evident in the gradual clogging of the Geruwa branch over time.

Meanwhile, the neighboring Koshi River has experienced declining water discharges since 2000, largely due to dam construction, and similar trends may unfold in the Karnali basin as hydropower projects increase in the future.

*Data availability.* Discharge time series for Chisapani (1962-2019) and Chatara (1977-2015) were obtained from the Department of Hydrology and Meteorology in Kathmandu, Nepal. Locations of hydropower stations with their license status is available on the website of the Department of Electricity Development (DoED), Nepal. In particular, data for hydropower stations with a survey license has been obtained from http://www.doed.gov.np/license/13, those with a construction license from http://www.doed.gov.np/license/66, and with an operation license (operating power plants) from http://www.doed.gov.np/license/54. DEM data were downloaded from https://opentopography.org. Landsat images were obtained from the Google Earth Engine repository. Historical maps were obtained from the David Rumsey Map Collection (https://www.davidrumsey.com).

*Author contributions.* KG, TB, and AB jointly conceptualized and designed the study. KG and TB planned the field campaigns. KG and MW collected the field data. KG and MR performed the data analysis. KG and AB drafted the manuscript. All authors reviewed, edited, and approved the final version. AB and TB secured the funding and provided overall supervision.



*Competing interests.* The authors declare that they have no conflict of interest.

*Acknowledgements.* We acknowledge Mo de Jong for her assistance in acquiring the field data. We thank staff of the National Trust for Nature Conservation (NTNC) Bardiya Conservation Program and Bardiya National Park, Punaram Chaudhary and Omkar Tharu, for their logistic support during fieldwork. The authors thank the College of Engineering and Management, Nepalgunj, for providing laboratory resources.

*Financial support.* This research project is part of the NWA-funded research program 'Save the tiger! Save the grasslands! Save the water!' (https://savethetiger.nl, grant no. NWA.1292.19.146), which is part of 'Research along Routes by Consortia' (NWA-ORC). The program is financed by the Dutch Research Council NWO with co-financing by four project partners (Planet, Rotterdam Zoo, Himalayan Tiger Foundation, and Practical Action).

275

280



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
