# Peer review of "Fluvial fans in the Himalayan Terai: A gradual shift of the Karnali River from double to single branch"

_EGUsphere, 2025_

## Author Comment (AC1)

**Dear reviewer, dear Jim,**

Thank you for your constructive feedback. We have worked on the two main comments you have pointed out. Our response is listed in blue.

The authors present a well-crafted, well-written investigation of recent changes to two fans of the Himalayan Terai region, with specific focus on the evolution of the Karnali River from a two-branch to a single-branch system. The topic is of considerable interest, and the manuscript is in pretty good shape – I found only a very few minor issues to comment on throughout.

Nonetheless, I think that the authors need to do a better job on two aspects of the study. First, their study would be clearer and make more sense to the reader if it was presented as a hypothesis-driven investigation. Second, the available data to test their leading hypothesis (which they present as a "finding" in the discussion) is quite limited, and I think they should explicitly acknowledge that additional work would be desirable to provide a more comprehensive test.

On the first point: the introduction reads as if the authors are going to just look at some data and see if anything interesting emerges. But I am sure that they had some specific ideas about why a fan system could evolve from a 2-channel to a single-channel system, and that these ideas guided their data collection. The paper would be much improved if specific hypotheses were offered at the outset, and the data collection effort was presented as an effort to test these hypotheses. As written, the author's main conclusion is presented first at the end of the Results section, which is really too late to introduce this. Better to see specific hypotheses at the beginning of the paper, and then explain which of these hypotheses are supported by the author's data.

On the second point: the only real support available that I could discern for the idea that one of the channels was "plugged" by coarse bed material in 2009 is the large grain size at the outside of a bend near the head of the fan. This seems like minimal support to me: ideally some evidence of the deposit itself might be available, or some other supporting evidence. I would like the authors to either more clearly summarize the existing evidence (in case I have missed something), and then discuss additional evidence would could (should?) be collected before accepting this working hypothesis (e.g., before and after topographic surveys, field mapping of the coarse deposit from 2009, and so on). This discussion would strengthen the scientific impact of the author's work.

These suggestions are important, and perhaps constitute a significant revision of the presentation, but should be easily accomplished and really only a minor effort for the authors. I have indicated major revisions, but I think they will be easily accomplished and the present manuscript greatly improved as a result.

Jim Pizzuto

Dept. of Earth Sciences (retired) University of Delaware, USA

**Response to Comment 1**

Observations on the gradually declining discharge in the eastern Geruwa branch on the fluvial fan of the Karnali river system have motivated this study. The research question that our study has aimed to answer is:

What are the causes of the flow partitioning at the river bifurcation on the Karnali fluvial fan to increasingly disfavor its eastern Geruwa branch since 2009?

To answer this question, we formulated the following sub-questions:

- A. How has the flow partitioning at the Chisapani bifurcation on the Karnali fluvial fan developed over the past centuries?
- B. How do the hydrogeomorphic characteristics of the Karnali fluvial fan compare to those of other fluvial fans in the Himalayan foothills, in particular the Koshi fluvial fan in eastern Nepal?\*
- C. Does the monsoon-dominated hydrograph at Chisapani (located right upstream of the bifurcation) show deviations from its regular variability?
- D. What role have water intakes had on the flow partitioning between the two branches of the fluvial fan?
- E. What role have the embankments along the two branches of the fluvial fan had on the flow partitioning?
- F. What role has the elevated land to the east of the Karnali River and downstream of the Chisapani bifurcation had on the flow partitioning?
- G. What can explain the *gradual* decline of the flow discharge into the Geruwa since the 2009 season?

\*We have chosen for a comparison of the Karnali and Koshi fluvial fans as the latter is one of the most studied systems in this region.

Our data collection has been guided by the above questions rather than by one hypothesis or a set of hypotheses. We will adjust the Introduction section to include this information.

**Response to Comment 2**

We agree on your point that images of the bed surface sediment provide limited evidence in support of our hypothesis. Unfortunately, our domain of interest is a protected and data scarce area. We do not have data from topographic surveys or surface sediment samples from the past.

Following your question, we have investigated remotely sensed data of the area in more detail. We have analyzed satellite images and global DEM data sets from before and after 2009. We have considered SRTM 30m Global DEM data, which represents earth surface elevation from 2000, as well as the composite Copernicus 30m Global DEM collected over the period 2011-2013. Most of the Copernicus data for the Karnali fluvial fan stems from 2011. These two DEM data sets are the only ones available to us.

In addition to DEM data, we have analyzed Landsat images from 2000, 2009 (before the monsoon season), and 2011 to extract land cover and stream network information, both under low discharge conditions at Chisapani (Figure 1).

In 2000, the Geruwa branch carried a significant portion of the water discharge, and under low flow conditions at least two channels supplied water to the eastern Geruwa branch from an outer bend at Dolphin Point in the upstream Karnali River (Figure 1A). By 2009 (yet before the double monsoon peak), a new eastern channel has formed near the eastern fan boundary (Figure 1B). In 2011, this eastern channel supplying water to the Geruwa branch ceased to exist (Figure 1C). Under low flow conditions, only one channel still supplies water to the Geruwa branch, and it has narrowed since

2009. As a result, a large part of the Chisapani water discharge is transported through the Kauriala branch.

The disappearance and decline of the eastern channels between 2009 and 2011 reflect the decline of river discharge into the Geruwa branch, which seems to be associated with sediment deposition in the channel taking off from the outer bend at Dolphin point.

We have computed the difference in surface elevation between the composite Copernicus DEM (2011-2013) and the SRTM DEM (2000) (Figure 1D-E). In 2011 the upstream zone of the Geruwa branch has become elevated compared to 2000 (Figure 1E). Our domain of interest is unvegetated, which implies that DEM surface elevation data reflecting canopy elevation for vegetated areas does not affect estimates. Sediment deposition between 2011-2013 and 2000 and the resulting elevation difference across the upstream region of the Geruwa branch seem to have restricted the water discharge into the Geruwa branch.

In addition, we have analyzed grain size distributions of the bed surface sediment. To this end, we have determined surface grain-size distributions from images taken at various locations along the fluvial fan (Figure 2). We have used Segmenteverygrain (Sylvester et al., 2025), a python-based tool, to determine surface sediment grain size from the images. The results indicate that the surface sediment across the outer bend at Dolphin Point and the upper reach of the Geruwa branch consists of a large amount of boulders compared to other locations.

The increase in bed level and boulder deposition across the upstream end of the Geruwa branch underline our hypothesis that a self-reinforcing mechanism was triggered (a) where boulders carried during peak flood discharge are deposited right downstream of the Karnali outer bend across the upstream end of the Geruwa branch; (b) deposited boulders, unable to move further downstream under these peak discharges, increase riverbed level across the upstream Geruwa branch, (c) this subsequently reduces the flow entering this branch; (d) this reduced flow limits the further transport of boulders into the Geruwa branch and (e) enhances further boulder deposition in the subsequent flood season.

We will include the above information in the manuscript.

Our response to detailed comments in blue:

**Comment 3.** Line 20. Should probably cite Figure 1 somewhere in this paragraph.

We will update the manuscript.

**Comment 4**. Line 51. In the paragraph above on anthropogenic stresses, land-use is not mentioned. Perhaps it should be included in the paragraph above, or, if it is not important here, not mentioned at all...???

Thank you, we will address aspects related to land use.

**Comment 5**. Line 67. Include: Section 6, which presumably interprets information from the previous sections and leads to the conclusions of Section 7.

We will refer to Sections 6 and 7.

**Comment 6**. Introduction, general comment. I think it would be helpful to present some specific hypotheses to test. This would provide context for the methods. Otherwise, why investigate flow

characteristics, dynamics, channel properties, and so on? Presumably these investigations are designed to answer specific questions derived from the authors' hypotheses....and the reader should know what these questions and hypotheses are.

We refer to our response to Comment 1.

**Comment 7**. Line 208. The hypothesis to be tested should be presented in the introduction, and specific methods available to test it should be outlined in the methods. Evidence from grain size only is somewhat weak, it seems to me. Is the hypothesis supported by topography analyses, or are these data insufficient or unavailable?

We refer to our response to Comments 1 and 2. We will implement the associated changes in the manuscript.

**Comment 8**. Line 215. But little direct evidence besides grain size has been presented for this. I think it remains a "working hypothesis", rather than a "finding". This may be semantics, but nonetheless important.

We refer to our response to Comment 2.

Figure 1: Wetted surface area indicating river channels across the Karnali fluvial fan in the years (A) 2000, (B) 2009, and (C) 2011. Data relate to low flow conditions: 300 m³/s in 2000, 304 m³/s in 200, and 296 m³/s in 2011. The inset at the top-left corner of each map shows the Dolphin point bend right upstream of the Geruwa branch, where the flow partitions between the Geruwa and Kauriala branches. (D) Difference in surface elevation across the Karnali fan between 2011-2014 (Copernicus DEM) and 2000 (SRTM DEM); (E) Inset of the elevation difference map at Dolphin point region or upstream end of the Geruwa branch with river channel outlines for the years 2000, 2009, and 2011.

Figure 2: Grain size distributions of the surface sediment obtained from image analysis at various locations across the Karnali fluvial fan. Only grains with a major axis length larger than 20mm could be distinguished.

**References**

Sylvester, Z., Stockli, D. F., Howes, N., Roberts, K., Malkowski, M. A., Poros, Z., Martindale, R. C., and Bai, W.: Segmenteverygrain: A Python module for segmentation of grains in images, Journal of Open Source Software, 10, 7953, https://doi.org/10.21105/joss.07953, 2025.

---

## Author Comment (AC2)

Dear reviewer,

Thank you for your constructive feedback. We have worked on the two main comments you have pointed out. Our response is listed in blue.

This manuscript investigates the gradual transition of the Karnali River fluvial fan system from a long-lived double-branch configuration to a single dominant channel. The study addresses an important and timely topic in fluvial geomorphology and river–fan dynamics, particularly in large Himalayan rivers where natural processes still dominate over engineering control. Overall, the manuscript is well written, and presents a possible explanation for Karnali River based on multiple datasets. However, I still have two major questions that need to be addressed.

1. The paragraphs are very short and lack clear logical flow, making the manuscript feel like a compilation of disconnected information. I suggest restructuring the paper around a coherent framework, such as scientific hypothesis → evidence → discussion → conclusion. Meanwhile, the Discussion section should be strengthened with more mechanistic explanations, as the current presentation of information is somewhat confusing.

2. The authors suggest that the river shift may be related to the 2009 monsoon season. Such as heavy 2009 monsoon season formed a sediment 'plug' in the upstream reach of the Geruwa branch. I think that this evidence appears relatively weak. Could the authors provide additional evidence to support this hypothesis?"

In addition to the above, there are some minor issues that I also suggest the authors consider.

Line 16-25. Could more recent literature be added here to highlight the significance of your study?

Line 40-51. At interannual timescales, human activities and climate change are the dominant drivers of river change; however, in certain regions or during extreme events, abrupt tectonic activity (such as earthquakes) and short-term sea-level fluctuations can also significantly influence river systems.

Line 61. Which natural factors are included here? These have not been specified earlier in the manuscript.

Line 143-144. This sentence should appear in the Discussion section.

Line 156-159 This is just a simple description; I would rather know more about the underlying mechanisms.

Line 187 "Sinuosity and Braiding Indices and Sinuosity Dynamics for". Should the first letter be capitalized?

Line 206-208 This is a good hypothesis, but how can it be tested? Is there any new evidence to support it?

Line 210-249 the Discussion section is overall too simplistic, merely ruling out some possible factors. I hope you can build on your results to provide more mechanistic insights.

Line 250-260. Could this be written as one or two paragraphs? There's no need for so many separate paragraphs.

**Response to Comment 1**

Observations on the gradually declining discharge in the eastern Geruwa branch on the fluvial fan of the Karnali river system have motivated this study. Following our response to Reviewer 1, we would like to emphasize that our research is exploratory and has been guided by the following questions rather than by a hypothesis. The research question that our study has aimed to answer is:

> What are the causes of the flow partitioning at the river bifurcation on the Karnali fluvial fan to increasingly disfavor its eastern Geruwa branch since 2009?

To answer this question, we formulated the following sub-questions:

A. How has the flow partitioning at the Chisapani bifurcation on the Karnali fluvial fan developed over the past centuries?
B. How do the hydrogeomorphic characteristics of the Karnali fluvial fan compare to those of other fluvial fans in the Himalayan foothills, in particular the Koshi fluvial fan in eastern Nepal?*
C. Does the monsoon-dominated hydrograph at Chisapani (located right upstream of the bifurcation) show deviations from its regular variability?
D. What role have water intakes had on the flow partitioning between the two branches of the fluvial fan?
E. What role have the embankments along the two branches of the fluvial fan had on the flow partitioning?
F. What role has the elevated land to the east of the Karnali River and downstream of the Chisapani bifurcation had on the flow partitioning?
G. What can explain the *gradual* decline of the flow discharge into the Geruwa since the 2009 season?

*We have chosen for a comparison of the Karnali and Koshi fluvial fans as the latter is one of the most studied systems in this region.

Following this reviewer and Reviewer 1's comments, we have done some additional analyses that provide more clarity and evidence to our concluding hypothesis. We refer to our response to Comment 2 below.

We will adjust the Introduction section to include this information and make the paper more coherent.

**Response to Comment 2**

We agree on your point that evidence in support of our hypothesis has limitations. This point was also raised by the first reviewer.

Following your and the first reviewer's suggestions, we have investigated remotely sensed data of the area in more detail. We have analyzed satellite images and global DEM data sets from before and after 2009. We have considered SRTM 30m Global DEM data, which represents earth surface elevation from 2000, as well as the composite Copernicus 30m Global DEM collected over the period 2011-2013. Most of the Copernicus data for the Karnali fluvial fan stems from 2011. These two DEM data sets are the only ones available to us.

In addition to DEM data, we have analyzed Landsat images from 2000, 2009 (before the monsoon season), and 2011 to extract land cover and stream network information, both under low discharge conditions at Chisapani (Figure 1).

In 2000, the Geruwa branch carried a significant portion of the water discharge, and under low flow conditions at least two channels supplied water to the eastern Geruwa branch from an outer bend at Dolphin Point in the upstream Karnali River (Figure 1A). By 2009 (yet before the double monsoon peak), a new eastern channel has formed near the eastern fan boundary (Figure 1B). In 2011, this

eastern channel supplying water to the Geruwa branch ceased to exist (Figure 1C). Under low flow conditions, only one channel still supplies water to the Geruwa branch, and it has narrowed since 2009. As a result, a large part of the Chisapani water discharge is transported through the Kauriala branch.

The disappearance and decline of the eastern channels between 2009 and 2011 reflect the decline of river discharge into the Geruwa branch, which seems to be associated with sediment deposition in the channel taking off from the outer bend at Dolphin point.

We have computed the difference in surface elevation between the composite Copernicus DEM (2011-2013) and the SRTM DEM (2000) (Figure 1D-E). In 2011 the upstream zone of the Geruwa branch has become elevated compared to 2000 (Figure 1E). Our domain of interest is unvegetated, which implies that DEM surface elevation data reflecting canopy elevation for vegetated areas does not affect estimates. Sediment deposition between 2011-2013 and 2000 and the resulting elevation difference across the upstream region of the Geruwa branch seem to have restricted the water discharge into the Geruwa branch.

In addition, we have analyzed grain size distributions of the bed surface sediment. To this end, we have determined surface grain-size distributions from images taken at various locations along the fluvial fan (Figure 2). We have used Segmenteverygrain (Sylvester et al., 2025) , a python-based tool, to determine surface sediment grain size from the images. The results indicate that the surface sediment across the outer bend at Dolphin Point and the upper reach of the Geruwa branch consists of a large amount of boulders compared to other locations.

The increase in bed level and boulder deposition across the upstream end of the Geruwa branch underline our hypothesis that a self-reinforcing mechanism was triggered (a) where boulders carried during peak flood discharge are deposited right downstream of the Karnali outer bend across the upstream end of the Geruwa branch; (b) deposited boulders, unable to move further downstream under these peak discharges, increase riverbed level across the upstream Geruwa branch, (c) this subsequently reduces the flow entering this branch; (d) this reduced flow limits the further transport of boulders into the Geruwa branch and (e) enhances further boulder deposition in the subsequent flood season.

We will include the above information in the manuscript.

Our response to additional comments:

**Comment 3**. Line 16-25. Could more recent literature be added here to highlight the significance of your study?

We will  include relevant literature in the updated manuscript.

**Comment 4**. Line 40-51. At interannual timescales, human activities and climate change are the dominant drivers of river change; however, in certain regions or during extreme events, abrupt tectonic activity (such as earthquakes) and short-term sea-level fluctuations can also significantly influence river systems.

Thank you for your comment. We agree that human activities, climate change, and also extreme events influence river morphodynamics. We will add this information in the introduction.

**Comment 5**. Line 61. Which natural factors are included here? These have not been specified earlier in the manuscript.

The natural factors include hydrology, topography and geology. We will update the manuscript.

**Comment 6**. Line 143-144. This sentence should appear in the Discussion section.

Agreed, we will update the manuscript.

**Comment 7**. Line 156-159 This is just a simple description; I would rather know more about the underlying mechanisms.

Indeed, the explanation on why bifurcation stability is not a closed topic was brief. We will expand the manuscript and explain remaining challenges more clearly:

1. Nonlinear morphodynamic feedbacks

The stability of river bifurcations is controlled by strongly nonlinear feedbacks between flow partitioning, sediment transport, and channel bed level. Perturbations in discharge or sediment supply can trigger self-reinforcing adjustments (e.g. preferential channel deepening), making bifurcation behavior sensitive to boundary conditions and initial states (Blom et al., 2024; Bolla Pittaluga et al., 2003; Wang et al., 1995).

2. Scale dependence and long adjustment times

Bifurcation dynamics emerge from processes acting across multiple spatial and temporal scales, from bar-scale sediment sorting to basin-scale sediment supply and base-level change. It remains difficult to distinguish transient responses from long-term stable configurations as bed level adjustments often span decades to centuries (Bolla Pittaluga et al., 2015; Kleinhans et al., 2008).

3. Anthropogenic modification of bifurcation dynamics

Engineering interventions such as dams, groynes, dredging, and bank fixation strongly modify flow and sediment regimes at bifurcations. Consequently, observed stability may reflect active management rather than intrinsic morphodynamic behavior, complicating the interpretation of stability in both natural and regulated river systems (Kleinhans et al., 2013; Mendoza et al., 2019).

**Comment 8**. Line 187 "Sinuosity and Braiding Indices and Sinuosity Dynamics for". Should the first letter be capitalized?

Not necessarily, yet this is done to clarify that we use the first letters of the indices also as acronyms.

**Comment 9**. Line 206-208 This is a good hypothesis, but how can it be tested? Is there any new evidence to support it?

Thank you for your comment. We refer to our response to Comment 2 for extra analyses providing additional evidence. In future research we expect that numerical modelling may shed added light on the mechanism and the validity of the hypothesis. Nevertheless, this is a data-scarce environment and data-scarce case, which would hinder researchers in deciding whether the numerically modelled case sufficiently replicates the real world case.

**Comment 10**. Line 210-249 the Discussion section is overall too simplistic, merely ruling out some possible factors. I hope you can build on your results to provide more mechanistic insights.

We have done additional analyses to provide more evidence to support our hypothesis. We refer to our response to Comment 2, and we will update the manuscript accordingly.

**Comment 11**. Line 250-260. Could this be written as one or two paragraphs? There's no need for so many separate paragraphs.

We will update the manuscript.

[Figure]

Figure 1: Wetted surface area indicating river channels across the Karnali fluvial fan in the years (A) 2000, (B) 2009, and (C) 2011. Data relate to low flow conditions: 300 m³/s in 2000, 304 m³/s in 200, and 296 m³/s in 2011. The inset at the top-left corner of each map shows the Dolphin point bend right upstream of the Geruwa branch, where the flow partitions between the Geruwa and Kauriala branches. (D) Difference in surface elevation across the Karnali fan between 2011-2014 (Copernicus DEM) and 2000 (SRTM DEM); (E) Inset of the elevation difference map at Dolphin point region or upstream end of the Geruwa branch with river channel outlines for the years 2000, 2009, and 2011.

[Figure]

Figure 2: Grain size distributions of the surface sediment obtained from image analysis at various locations across the Karnali fluvial fan. Only grains with a major axis length larger than 20mm could be distinguished.

**References**

Blom, A., Ylla Arbós, C., Chowdhury, M. K., Doelman, A., Rietkerk, M., and Schielen, R. M. J.: Indications of ongoing noise-tipping of a bifurcating river system, Geophysical Research Letters, 51, e2024GL111846, https://doi.org/10.1029/2024GL111846, 2024.

Bolla Pittaluga, M., Repetto, R., and Tubino, M.: Channel bifurcation in braided rivers: Equilibrium configurations and stability, Water Resources Research, 39, https://doi.org/10.1029/2001WR001112, 2003.

Bolla Pittaluga, M., Coco, G., and Kleinhans, M. G.: A unified framework for stability of channel bifurcations in gravel and sand fluvial systems, Geophysical Research Letters, 42, 7521–7536, https://doi.org/10.1002/2015GL065175, 2015.

Kleinhans, M. G., Jagers, H. R. A., Mosselman, E., and Sloff, C. J.: Bifurcation dynamics and avulsion duration in meandering rivers by one-dimensional and three-dimensional models, Water Resources Research, 44, https://doi.org/10.1029/2007WR005912, 2008.

Kleinhans, M. G., Ferguson, R. I., Lane, S. N., and Hardy, R. J.: Splitting rivers at their seams: bifurcations and avulsion, Earth Surface Processes and Landforms, 38, 47–61, https://doi.org/10.1002/esp.3268, 2013.

Mendoza, A., Soto-Cortes, G., Priego-Hernandez, G., and Rivera-Trejo, F.: Historical description of the morphology and hydraulic behavior of a bifurcation in the lowlands of the Grijalva River Basin, Mexico, CATENA, 176, 343–351, https://doi.org/10.1016/j.catena.2019.01.033, 2019.

Sylvester, Z., Stockli, D. F., Howes, N., Roberts, K., Malkowski, M. A., Poros, Z., Martindale, R. C., and Bai, W.: Segmenteverygrain: A Python module for segmentation of grains in images, Journal of Open Source Software, 10, 7953, https://doi.org/10.21105/joss.07953, 2025.

Wang, Z. B., De Vries, M., Fokkink, R. J., and Langerak, A.: Stability of river bifurcations in ID morphodynamic models, Journal of Hydraulic Research, 33, 739–750, https://doi.org/10.1080/00221689509498549, 1995.

---

## Author Comment (AC3)

Dear reviewer, dear Ivo,

We sincerely appreciate the feedback and thoughtful suggestions. Addressing the points you raised has helped us clarify our arguments and strengthen the paper. In particular, we have addressed the concern regarding the claim that the Karnali River is transitioning from a multi-branch to a single-branch system following the 2009 monsoon. Our response is highlighted in blue.

The article investigates the evolution of the Karnali fan over the past decades using data from historical surveys and remote sensing imagery. The authors discuss the implications of the fan evolution on ecosystems, which is a novel and important aspect. Furthermore, the authors present the first article (that I am aware of), which uses historical maps of the Karnali fan dating back to the 18th century, and hence adds to our understanding of the fan.

Main criticism

I have one major concern regarding the claim that the Karnali River transitioned from a multi-branch river to a single-branch river following the 2009 monsoon season. This statement is repeated from the title to the conclusion, but I cannot see sufficient evidence to support this transition. I am not convinced that the Karnali is a single-branch river for several reasons:

- Satellite images after 2009 show that the Eastern branch still carries water and is, thus, still an active channel. Please see the attached images of Oct. 2014-2021 as an example.

- Figure 6a shows that the Western branch had a lower water surface area between 2000-2009 than the Eastern branch between 2010-2020. It is not logical to argue that it transitioned to a single-branch river after 2009, when the low-flow partitioning is more balanced after 2009 than before.

- The analysis is based on imagery during low-flow conditions. Dingle (https://doi.org/10.1130/G46909.1 supplements) shows that normal to high flows overflow the plug, and that the Eastern branch still receives water during low flows despite the plug. Hence, the Eastern branch is active, and low flow conditions are insufficient to determine the activity of a branch.

Minor aspects

- Branch characteristics: The Eastern and Western branches have different channel characteristics, whereas the Eastern branch is composed of more but narrower braid channels. These narrower channels are more difficult to classify from 30m satellite imagery, which likely leads to an underestimation of the water surface area in the Eastern branch. It may be worth addressing this limitation in the discussion.

- Potential future evolution: The authors argue that with human interventions, the likelihood of future channel switches decreases (lines 245ff). I believe that this section needs more context. Dingle, 2017 (doi:10.1038/nature22039) shows that the gravel supply to the fan originates from the Siwaliks. The fan will not be blocked from its gravel supply unless a hydropower station is constructed at the mountain gauge. Upstream stations may reduce the flow rate (and hence transport capacity), which may lead to a decrease in gravel supply. However, increasing frequency and magnitude of intense rainfall may counteract and lead to increasing gravel supply (e.g. https://doi.org/10.3126/jalawaayu.v1i1.36448)

Additional thoughts:

- 2009 monsoon season: There is a book chapter on bifurcation switches in the Karnali fan by C. Cload called monsoon-driven changes to river bifurcations in Nepal (DOI: 10.1201/9781003475378-6), which may be of interest.

- It may be worth assessing the variation of water surface area with flow (e.g. comparing multiple images during a year). This may be beyond the scope of the study, but I believe it would add depth to the article because it provides evidence about how sensitive medium to high flows are to bifurcation changes. Without any consideration of medium to high flows in the analysis, it should be clearly stated that the findings relate to low-flow conditions only.

I disagree with the authors that the Karnali transitioned to a single-branch channel and feel that the narrative of the article needs to be adjusted. Nonetheless, I believe that it is a valuable article because it improves our understanding of the channel evolution during low flow conditions, which prevails for most of the year and is important for ecosystems and communities. I would further like to highlight that the figure presentation is excellent.

Ivo Pink

**Response to major comments**

We would like to emphasize that we have not intended to claim that the Karnali River over its fan has turned into a single-channel system. Rather, we argue that the system shows a gradual shift toward a system where one channel is dominant and that this shift is still ongoing. Please also note this nuance from the second part of the manuscript title: "A gradual shift of the Karnali River from double to single branch".

Our remote sensing analysis could indeed only consider low flow conditions because of cloud-cover during monsoon conditions. We are aware of and agree with the point that such analysis under low flow conditions has limitations and that extension to high flow conditions needs to be done with care. To support our arguments, we have analyzed newly available measured discharge data in detail.

In 2023, we installed two flow-depth monitoring stations in the downstream reaches of the Kauriala and Geruwa branches, approximately 30 km downstream of Chisapani. We have used these flow depth data to determine the daily discharge in the two branches over the period between November 2023 and March 2025. In addition, we analyzed Karnali discharge data at Chisapani (located right upstream of the Karnali bifurcation) over the same period. These water discharge data at Chisapani became available to us through the Department of Hydrology and Meteorology, Nepal. We analyzed the ratio of the discharge in the two branches as a function of the discharge at Chisapani. We found that the water share in the Geruwa branch ranges from 5% under low flow conditions to 20% under high flow conditions (Figure 1). These data cover high flow as well as low flow conditions.

Dingle et al. (2020) observed a value of approximately 50% of the Karnali discharge going into the Geruwa branch during a moderate monsoon discharge of about 4500 $m^3$/s at Chisapani in August 2017. The analysis described above shows, for similar discharges, that this share has decreased to about 20% in the period 2023-2025 (Figure1). This indicates that the Geruwa branch now receives a smaller share of water than it used to in 2017. This supports the fact that the Geruwa discharge is declining not only under low flows but also higher flows. We emphasize that we have no information on temporal changes in discharge partitioning between the branches under more extreme monsoon conditions, and we will stress this in the revised manuscript.

Our other arguments are of a more qualitative nature, and relate to data on landcover and vegetation for the Geruwa floodplain. These data indicate a temporal reduction of floodplain surface sediment dynamics in the Geruwa branch. Bijlmakers et al.(2023) show that higher peak discharge at Chisapani corresponded to greater vegetation removal in the Geruwa floodplain until 2009. This correlation of vegetation removal with water discharge indicates that grassland cover may be used as a proxy for discharge through the extent of floodplain surface sediment dynamics . Since 2009 grassland area in the Geruwa floodplain has increased with time at the expense of bare substrate and water cover. We attribute this expansion of grassland cover to the temporal reduction of hydromorphodynamic activity (i.e., surface sediment reworking) in the Geruwa floodplain. We have

observed this temporal increase in grassland cover during our field monitoring campaigns. Moreover, these observations have been confirmed in interviews with employees of the Bardia National Park authority and local residents. They have reported a temporal decline in flooded areas and increase of vegetation cover in the Geruwa floodplain.

We would like to emphasize also that the Geruwa branch no longer receives water during low flow conditions naturally. While this may have been the case until 2021, since 2022 Geruwa discharge during low flows results from human excavation activities. Without such intervention, the sediment plug would prevent any flow into the Geruwa during low flows.

We will add the above extra analyses to the manuscript, and will expand the discussion accordingly.

[Figure]

*Figure 1: Ratio of discharge of Kauriala to Geruwa plotted against the Chisapani discharge shows the variation in discharge partitioning between the two branches range from about 5% at low flows to about 20% under high flow conditions. Water discharge in the downstream branches has been computed from time series of measured flow depth at Sattighat bridge (Kauriala) and Kothiyaghat bridge (Geruwa), both approximately 30 km downstream Chisapani, measured between 7 Nov 2023 to 15 March 2025. Chisapani water discharge over the same period is determined from flow depth data provided by DHM, Nepal. Data includes 2024 monsoon cycle with a peak discharge close to 10,000 m³/s.*

**Response to minor comments:**

While updating the manuscript we shall keep in mind the request for a more detailed description of the limitations or difficulties in the interpretation of satellite images for narrow channels, as well as the notes on human interventions.

**References**

Bijlmakers, J., Griffioen, J., and Karssenberg, D.: Environmental drivers of spatio-temporal dynamics in floodplain vegetation: grasslands as habitat for megafauna in Bardia National Park (Nepal), Biogeosciences, 20, 1113–1144, https://doi.org/10.5194/bg-20-1113-2023, 2023.

Dingle, E. H., Sinclair, H. D., Venditti, J. G., Attal, M., Kinnaird, T. C., Creed, M., Quick, L., Nittrouer, J. A., and Gautam, D.: Sediment dynamics across gravel-sand transitions: Implications for river stability and floodplain recycling, Geology, 48, 468–472, https://doi.org/10.1130/G46909.1, 2020.